# Evaluation on Biocontrol Efficacy of *Episyrphus balteatus* De Geer (Diptera: Syrphidae) Against *Aphis craccivora*, *Myzus persicae,* and *Megoura crassicauda*

**DOI:** 10.3390/insects16080774

**Published:** 2025-07-28

**Authors:** Shanshan Jiang, Hui Li, Kongming Wu

**Affiliations:** 1State Key Laboratory for Biology of Plant Diseases and Insect Pests, Institute of Plant Protection, Chinese Academy of Agricultural Sciences, Beijing 100193, China; jiangss97@163.com (S.J.); lihuilh521@163.com (H.L.); 2Department of Entomology, China Agricultural University, Beijing 100193, China; 3National Center of Technology Innovation for Comprehensive Utilization of Saline-Alkali Land, Dongying 257300, China

**Keywords:** *Episyrphus balteatus*, aphid, functional response, biocontrol efficacy, preference

## Abstract

*Episyrphus balteatus* is the only one hoverfly species among the top 25 biological control agents by adoption across countries. This study systematically evaluated the predatory functional responses, control efficacy, and oviposition and predatory preferences of *E. balteatus* against *Aphis craccivora* Koch, *Myzus persicae* Sulzer, and *Megoura crassicauda* Mordvilko under laboratory conditions. The results showed that the best functional response model of both second- and third-instar *E. balteatus* larvae to these aphid species was the Holling type III model, except for the third instar to *A. craccivora*, for which the Holling type II model was superior. *E. balteatus* had good biocontrol efficiency, and the oviposition and predatory preferences of *E. balteatus* were consistent with preferring *M. crassicauda*. These results strengthened and optimized the application of *E. balteatus* as BCAs for these three aphid species.

## 1. Introduction

Hoverflies (Syrphidae) constitute a major group within the order Diptera, encompassing approximately 200 genera and nearly 6000 species. They exhibit a global distribution and are found on all continents, except some remote islands [1,2]. The family Syrphidae comprises three subfamilies: Syrphinae, Eristalinae, and Microdontinae. Among these, only species within the subfamily Syrphinae (representing approximately one-third of hoverfly diversity) are predatory. Their larvae feed on pests such as whiteflies, thrips, and scale insects, with a particularly significant role in suppressing aphid populations [3]. For instance, a single third-instar larva can consume up to 60 cabbage aphids (*Brevicoryne brassicae*) per day, with lifetime consumption exceeding 300 aphids [4]. And adult hoverflies preferentially oviposit within aphid colonies [5], ensuring that emerging larvae can promptly regulate aphid numbers [6]. *Episyrphus balteatus* De Geer (Diptera: Syrphidae), commonly known as the marmalade hoverfly, is a holometabolous insect that undergoes four developmental stages of ontogeny: egg, larva, pupa, and adult [7]. Its distribution ranges from Asia and Europe to Africa and Australia, with a particularly dominant presence in China [8]. The larvae of *E. balteatus* are predators with a broad diet, feeding on over 100 species of aphids and some young lepidopteran larvae (such as those of *Spodoptera frugiperda* Smith) worldwide [9,10], performing an effective biological control function against pests [11]. Biological control efficacy studies demonstrate that releasing third-instar *E. balteatus* larvae against soybean aphids at a predator/prey ratio of 1:150 achieved a pest suppression rate of 59.87% after 72 h [12]. Similarly, releasing newly hatched *E. balteatus* larvae for aphid control in greenhouses on vegetables and chrysanthemums resulted in aphid population reduction rates exceeding 80% after 72 h [13]. As an important migratory hoverfly species, it can seasonally migrate over long distances, consuming trillions of aphids, due to which it represents the only syrphid species among the 25 most important biological control agents (BCAs) used in augmentative biological control [14], thereby playing a crucial role in maintaining the stability of global agricultural ecosystems [3,15]. Aphids (Hemiptera: Aphididae) are some of the most economically significant agricultural pests globally. They cause direct damage by feeding on the plant phloem and indirect damage by transmitting plant viruses and producing honeydew, which hinders photosynthesis [16]. *Aphis craccivora* Koch, *Myzus persicae* Sulzer, and *Megoura crassicauda* Matsumura are three common aphid species in global agricultural ecosystems within the Hemiptera order and Aphididae family. *A. craccivora* is an economically important pest that affects legumes (such as peas, cowpeas, and peanuts) worldwide. Notably, *A. craccivora* can transmit two major plant viruses, bean leaf roll virus and faba bean necrotic yellows virus, which severely impact the quality and yield of legumes [17]. *M. crassicauda* is another globally significant pest that primarily harms leguminous plants such as broad beans, peas, and soybeans, among which broad beans are the most severely affected [18]. *M. persicae*, a cosmopolitan pest, has a broad host range covering more than 30 families and over 100 plant species. More critically, it transmits more than 100 plant viruses, such as beet yellows virus and potato viruses [19]. Additionally, the strong reproductive capacity, ecological adaptability, and pesticide resistance of these aphid species have led to increasingly severe damage to crops, posing a great threat to the balance of the ecosystem. In aphid control, the use of sustainable methods such as biological control can help address the environmental risks and the ‘3R’ problems (resistance, resurgence, and residue) associated with reliance on insecticides [20]. Among these methods, utilizing natural enemy insects to enhance biological control services has emerged as a promising alternative.

The natural enemies of aphids include parasitic natural enemies, predatory natural enemies, pathogenic microorganisms, etc. [21]. Predatory natural enemies such as *E. balteatus* are generalist species, targeting a wide host range, which gives them great potential as BCAs against insect pests compared to specialist species like parasitoids, which exhibit a narrow host range [22]. However, parasitoids still dominate approximately 80% of biocontrol programs due to their lower food requirement and ability to maintain balance at lower host densities [23]. To enhance the use of predators as generalist BCAs, a deeper understanding of their predatory capabilities, control efficacy, and preferences for prey is one of the approaches that can promote their use in applied biological control practices.

This study aims to systematically investigate the predatory ability, control efficacy, and oviposition and predatory preferences of *E. balteatus* against *A. craccivora*, *M. persicae*, and *M. crassicauda* through controlled laboratory experiments. The specific objectives include (1) evaluating the predatory functional response of *E. balteatus* toward the three aphid species; (2) assessing the control efficacy of *E. balteatus* against the three aphid species under caged conditions; and (3) exploring its oviposition and predatory preferences for aphids. Through this research, we hope to strengthen and optimize the application of *E. balteatus* to these three aphid species as a BCA.

## 2. Materials and Methods

### 2.1. Test Aphid and Hoverfly

Aphids (*A. craccivora*, *M. persicae*, and *M. crassicauda*) and *E. balteatus* were collected from an experimental field at the Langfang Experimental Station, Chinese Academy of Agricultural Sciences (Langfang, China; 39°30′29′′ N, 116°36′8′′ E). They were reared in a greenhouse at 23 ± 1 °C, 50 ± 5% RH, and 16:8 (L:D) h. The three aphid species were reared on broad bean plantlets. The broad bean plantlets were planted in plastic pots (12 cm upper diameter × 9 cm lower diameter × 10 cm height) filled with a substrate of nutrient soil and vermiculite. The captured *E. balteatus* adults (17 females, 20 males) were placed separately in nylon mesh cages (80 cm × 80 cm × 80 cm). Inside the cages, they were provided with a mixture of pollen (rape/corn = 3:1) and 10% *v*/*v* honey solution for feeding, as well as broad bean plantlets infested with aphids for oviposition. The honey solution was replaced once a day, and the pollen and broad bean plantlets were replaced every four days. The larvae hatched from the eggs were reared on a mixed diet of aphids in plastic containers (50 cm × 40 cm × 15 cm) until they pupated.

### 2.2. Predatory Functional Response of E. balteatus to Aphids

One *E. balteatus* larva that was starved for 24 h was introduced into a Petri dish (with small ventilation holes in the lid, 9 cm diameter × 1.5 cm height) that contained broad bean leaves (6 pcs/dish). Mature female aphids of similar body sizes (*A. craccivora*, *M. persicae*, or *M. crassicauda*) were placed into the Petri dishes. And the number of consumed aphids, determined by counting either the aphid exoskeletons or the remaining live aphids over a 24 h period, was recorded. Each treatment of predator–prey combination (a second or third instar *E. balteatus* larva to each density of each aphid species) was repeated five times. The aphid density settings are shown in Table 1.

### 2.3. Control Efficacy of E. balteatus to Aphids Under Caged Conditions

One ten-day-old female adult of *E. balteatus* that was starting to lay eggs was reared in a cage (50 cm × 35 cm × 45 cm, 200-mesh nylon) with broad bean plantlets (3 pots per cage, 20 plants per pot) infested with aphids (*A. craccivora*, *M. persicae*, or *M. crassicauda*), mixture of pollen (rape/corn = 3:1), and 10% *v*/*v* honey water. The initial aphid density was set at five gradients (Table 2); each treatment was repeated 3 times. The number of aphids and larvae in the cages was recorded on the 3rd, 6th, 9th, and 12th days, and the aphid population decline rate was calculated using the following equation: Population decline rate (%) = [(No. of pre-treatment aphid population − No. of post-treatment aphid population)/No. of pre-treatment aphid population] × 100.

### 2.4. Preference of E. balteatus for A. craccivora, M. persicae, and M. crassicauda

#### 2.4.1. Oviposition Preference

Ten-day-old adult *E. balteatus* individuals (5 females and 5 males) that had already begun laying eggs were placed in a nylon mesh cage (80 cm × 80 cm × 80 cm). Three pots of broad bean plantlets, each infested with *A. craccivora*, *M. persicae*, or *M. crassicauda* (200 aphids per plantlet, 5 plantlets per pot), were arranged at equal intervals along the diagonal of the cage for oviposition. Additionally, a mixture of pollen (rape/corn = 3:1) and 10% *v*/*v* honey water was provided for feeding. The number of *E. balteatus* eggs on the broad bean plantlets with different aphid species was recorded daily. The experiment was repeated five times. The oviposition preference rate (%) = (number of eggs on a type of aphid plantlet/total number of eggs) × 100.

#### 2.4.2. Predatory Preference

The method was the same as described in Section 2.2. Adult aphids (*A. craccivora*:*M. persicae*:*M. crassicauda* = 1:1:1) were attached to the Petri dish; for 2nd-instar larvae, 10 individuals of each aphid species were added, and for 3rd-instar larvae, 30 individuals of each aphid species were added. After 24 h, the number of each aphid species consumed was recorded. Each treatment was repeated three times. The predatory preference (*Ci*) is calculated using the following equation [24]:*Ci* = (*Qi* − *Fi*)/(*Qi* + *Fi*),
where *Fi* represents the proportion of the *i-*th prey in the environment (the *Fi* value is constant at 1/3 in this experiment), and *Qi* represents the proportion of the *i*-th prey consumed by the predator. When *Ci* = 0, it indicates that the predator has no preference for the *i*-th prey species; when 0 < *Ci* < 1, it indicates a positive preference for the *i*-th prey species; and when −1 < *Ci* < 0, it indicates a negative preference for the *i*-th prey species.

### 2.5. Statistical Methodology

According to Okuyama and Ruyle [25], different functional response models can be described in general form using the equations in Table 3.

## 3. Results

### 3.1. Predatory Functional Response of E. balteatus to Aphids

The functional response of the second- and third-instar *E. balteatus* larvae to *A. craccivora*, *M. persicae*, and *M. crassicauda* is depicted in Figure 1. In the case of the second-instar *E. balteatus* larvae to these three aphid species, the predatory response curves generated by the Holling type II and Beddington–DeAngelis models and by the Holling type III and the θ-sigmoid models closely resemble each other. The third-instar larvae to *A. craccivora*, however, showed a similar fit between the θ-sigmoid and Beddington–DeAngelis functional response models (Figure 1; Table 4).

The estimated values for the functional response model parameters of the second- and third-instar *E. balteatus* larvae to *A. craccivora*, *M. persicae*, and *M. crassicauda* are presented in Table 4. According to the model selection method described by Hilborn and Mangel [27], which states that a smaller AIC value indicates a better-fitting model, the best functional response models of both second- and third-instar *E. balteatus* larvae to these three aphid species were the Holling type III model, except for the third-instar larvae to *A. craccivora*, for which the Holling type II model was better (Table 4).

### 3.2. Control Efficacy of E. balteatus to Aphids Under Caged Conditions

For the ratios of 1:500, 1:1000, 1:2000, and 1:4000, the *A. craccivora* population peaked on day 6 after inoculation, reaching 1553.33, 3033.33, 14,193.33, and 9396.67 individuals, respectively (except for the 1:6000 ratio, which peaked on day 9 at 11,500 individuals). As the larvae emerged and their numbers increased, the *A. craccivora* population decline rates for ratios of 1:500, 1:1000, 1:2000, 1:4000, and 1:6000 were 94.67%, 100.00%, 77.50%, 35.58%, and 24.72% on day 12, respectively, showing significant differences between the treatment groups (*F*_5,12_ = 213.431, *p* < 0.001) (Figure 2a,b, Table 5).

At hoverfly/aphid ratios of 1:2000, 1:4000, 1:6000, and 1:10,000, the *M. persicae* population peaked on day 6, reaching 3426.67, 5460, 8400, and 12,723.33 individuals, respectively (except for the 1:8000 ratio, which peaked earlier on day 3 at 10,043.33 individuals). Following larval emergence, the *M. persicae* population decline rates for ratios of 1:2000, 1:4000, 1:6000, 1:8000, and 1:10,000 reached 96.67%, 95.42%, 60.72%, 40.25%, and 21.10% by day 12, with significant differences between the treatment groups (*F*_5,12_ = 384.062, *p* < 0.001) (Figure 2c,d, Table 5).

At hoverfly/aphid ratios of 1:250, 1:500, and 1:1000, the *M. crassicauda* population exhibited continuous growth in the early stage, peaking on day 6 at 443.33, 1410, and 2543.33 individuals, respectively. In contrast, populations with ratios of 1:1500 and 1:2000 peaked later, on day 9, reaching 3433.33 and 4573.33 individuals. The aphid population at a ratio of 1:250 was completely eliminated by day 9, achieving a 100.00% population decline rate, a significantly higher efficacy than other ratios (*F*_5,12_ = 28,334, *p <* 0.001). At ratios of 1:500, 1:1000, 1:1500, and 1:2000, the population decline rates by day 12 were 84.67%, 40.33%, −98.22%, and −98.17%, respectively, with significant differences observed (*F*_5,12_ = 107.871, *p <* 0.001) (Figure 2e,f, Table 5).

### 3.3. Preference of E. balteatus for A. craccivora, M. persicae, and M. crassicauda

#### 3.3.1. Oviposition Preference

There were significant differences in the oviposition preference of *E. balteatus* adults among the three aphid species (*F*_2,12_ = 67.768, *p <* 0.001) (Figure 3). The oviposition preference rate was ranked as *M. crassicauda* (51.92%) > *A. craccivora* (28.32%) > *M. persicae* (19.76%).

#### 3.3.2. Predatory Preference

There were significant differences in the *Ci* of second-instar (*F*_2,6_ = 29.749, *p* = 0.001) and third-instar (*F*_2,6_ = 297.473, *p <* 0.001) *E. balteatus* larvae among the three aphid species (Table 6). The *Ci* was ranked as *A. craccivora* (*Ci >* 0) > *M. crassicauda* (*Ci >* 0) > *M. persicae* (*Ci <* 0).

## 4. Discussion

Numerous studies have documented the functional response of the cosmopolitan hoverfly *E. balteatus* [28,29]. However, data remain scarce regarding its predation on *A. craccivora*, *M. persicae*, and *M. crassicauda*. In this study, *E. balteatus* larvae demonstrated effective predation on all three aphid species, with third-instar larvae exhibiting greater voracity than second instars (under the highest prey density set in this experiment, the feeding amounts of third-instar larvae on *A. craccivora*, *M. persicae*, and *M. crassicauda* were 49.4, 56, and 28.8, respectively; for the second instar, they were 26, 39.4, and 13.8 (Figure 1)). These findings were also supported by Baskaran et al. [30], who documented the highest predation rates in third-instar larvae of four syrphid species on *Aphis gossypii*. Similar patterns were observed for *Pseudodoros clavatus* on *Aphis craccivora* [31] and for *E. balteatus upon Aphis fabae* [32]. The high predation of the final instar stage represents a logical consequence of both its larger body size and the additional nutritional demands required for subsequent pupal development. In our study, both second- and third-instar larvae of *E. balteatus* exhibited a Holling type III functional response to *Myzus persicae*, while it was Holling type II in Jalilian et al.’s study [33], which is likely attributable to geographic population differences. It is worth noting that the second- and third-instar *E. balteatus* to *A. craccivora* exhibited Holling type III and II functional responses, respectively, supporting the view that a single predator species can display variable responses depending on predator size/age and prey species/size [29]. The comparison of the a or h values obtained from different studies conducted and analyzed using a similar approach may be relevant [34]. According to the Holling type II model, the h values obtained for second-instar and third-instar *E. balteatus* larvae feeding on the three aphid species were consistently higher than those for *E. corollae* larvae (*A. craccivora*: L2 = 0.012 h, L3 = 0.007 h; *M. persicae*: L2 = 0.006 h, L3 = 0.005 h; *M. crassicauda*: L2 = 0.021 h, L3 = 0.011 h) [35]. Notably, second-instar *E. balteatus* larvae displayed higher a values on *A. craccivora* and *M. crassicauda* than third instars, a pattern consistent with observations by Evelin et al. [36] for *Allograpta exotica* preying on *A. craccivora*, where first-instar larvae exhibited higher attack rates than second instars.

The predator/prey release ratio serves as a crucial parameter in biological control applications. Our study demonstrated that the control efficacy of *E. balteatus* on these three aphid species showed a positive correlation with release ratios, consistent with Li et al.’s findings using *E. corollae* against *Aphis gossypii* [37]. While higher predator/prey ratios generally enhance control effectiveness under certain conditions, the recommended release ratios for *A. craccivora*, *M. persicae*, and *M. crassicauda*—considering cost-effectiveness—are 1:2000, 1:4000, and 1:500, respectively. However, *E. balteatus* exhibited significantly reduced or even ceased oviposition during later experimental stages, attributable to increasing aphid population density and aphid-induced wilting of broad bean plants, reflecting the mutual constraints of natural enemies and pest populations [38]. In this experiment, a 1:250 ratio of adult hoverflies to *M. crassicauda* achieved a 100.00% population decline rate by day 9, though no control effect was observed by day 3. The population decline rate of 1:150 ratio of third-instar *E. balteatus* larvae to soybean aphids was 59.87% by day 3 in Lan et al.’s study [12]. This demonstrates that adults (producing multiple larvae after about 3 days of egg hatching) offers broader and more effective control than larvae alone, albeit with a slower onset of action. In some scenarios, a combined application of both adults and larvae may be advisable for optimal results. Additionally, planting diverse flowering species or intercropping nectar-rich plants (such as rape, chrysanthemum, fatsia japonica, etc.) [39] near crops can enhance attraction of wild hoverfly adults for more sustainable biological control.

The ‘preference–performance’ hypothesis (also known as optimal oviposition theory) posits that female insects preferentially lay eggs on substrates that maximize larval survival and growth, which has been empirically supported by numerous studies [40,41]. *Episyrphus balteatus* adults demonstrated a significant oviposition preference for *M. crassicauda* in this study, and our previous studies showed that the developmental duration of *E. balteatus* larvae on *M. crassicauda* was significantly shorter than that on *A. craccivora* and *M. persicae*, and the larvae survival rate of *E. balteatus* on *M. crassicauda* was the highest [7], which echoes the above ‘preference–performance’ hypothesis.

The predatory preferences of hoverflies can significantly influence their effectiveness as BCAs against target pests [42]. Predatory preferences are influenced by multiple factors, including prey density, mobility, and the nutritional composition of prey (the primary determinant) [43]. According to the optimal foraging theory, predators generally select prey that best satisfies their survival and reproductive requirements [44]. This study revealed that both second- and third-instar *E. balteatus* larvae exhibited positive predatory preferences (*Ci* > 0) for *A. craccivora* and *M. crassicauda* under equal prey density conditions. This preference likely stems from the superior energy intake and enhanced developmental benefits provided by these two aphid species, as evidenced by significantly higher life-table parameters (net reproductive rate (*R*_0_), intrinsic rate of natural increase (*r*), and finite rate of increase (*λ*) compared to those fed on *M. persicae*) [7]. The oviposition preference and predatory preference (*Ci*) of *E. balteatus* for the three aphid species exhibited different ranking orders. This discrepancy can likely be attributed to the larger body size of adult *M. crassicauda* compared to *A. craccivora*, which could render them more difficult to prey upon.

Hoverflies are well known in the biological control of invertebrate pests, with studies confirming their efficacy in suppressing pest populations [45]. Currently, the most widely utilized and researched BCAs in commercial applications globally are parasitoid wasps, predatory flower bugs, and lacewings, etc., with only one hoverfly species—*E. balteatus*—ranking among the top 25 BCAs by adoption across countries [14]. This study systematically evaluates *E. balteatus*’s biocontrol efficacy against *A. craccivora*, *M. persicae*, and *M. crassicauda* under laboratory conditions, assessing its feasibility as a BCA for these aphid species. The complex and changeable natural environment, interspecific competition, and interference within hoverfly colonies merit attention. We advocate for expanded research on hoverfly biology, anticipating that such efforts could unlock their broader application in conservation and augmentative biological pest control.

## 5. Conclusions

The results showed that the best functional response model of both second- and third-instar *E. balteatus* larvae to these aphid species was the Holling type III model, except for the response of the third instar to *A. craccivora*, for which the Holling type II model was best. *E. balteatus* had good biocontrol efficiency, and the oviposition and predatory preferences of *E. balteatus* were consistent, with a preference for *M. crassicauda*. These results strengthen and optimize the application of *E. balteatus* as BCAs for these three aphid species.

## Figures and Tables

**Figure 1 insects-16-00774-f001:**
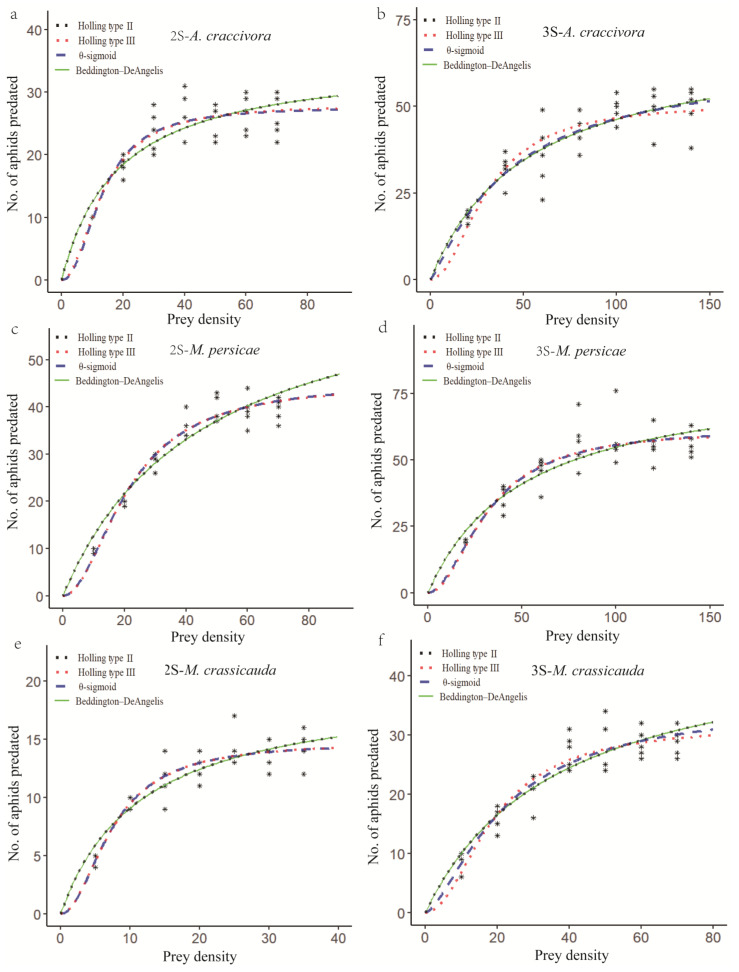
Functional response of 2nd- or 3rd-instar *E. balteatus* larvae to different densities of *A. craccivora* (**a**,**b**), *M. persicae* (**c**,**d**), or *M. crassicauda* (**e**,**f**). The symbols (*****) in the figure shows the number of aphids predated by *E. balteatus* larvae.

**Figure 2 insects-16-00774-f002:**
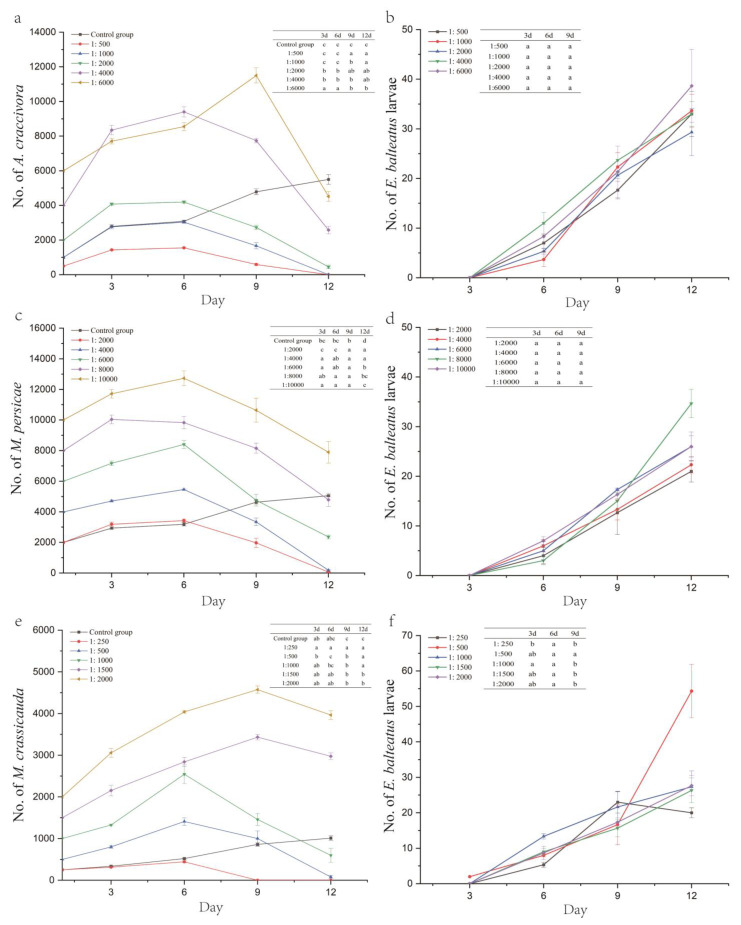
Number of aphids and *E. balteatus* larvae under different hoverfly/aphid release ratios (*E. balteatus–A. craccivora* (**a**,**b**), *E. balteatus–M. persicae* (**c**,**d**), and *E. balteatus–M. crassicauda* (**e**,**f**)). Different lowercase letters indicate statistically significant differences in no. of *E. balteatus* larvae (**b**,**d**,**f**) between treatment groups.

**Figure 3 insects-16-00774-f003:**
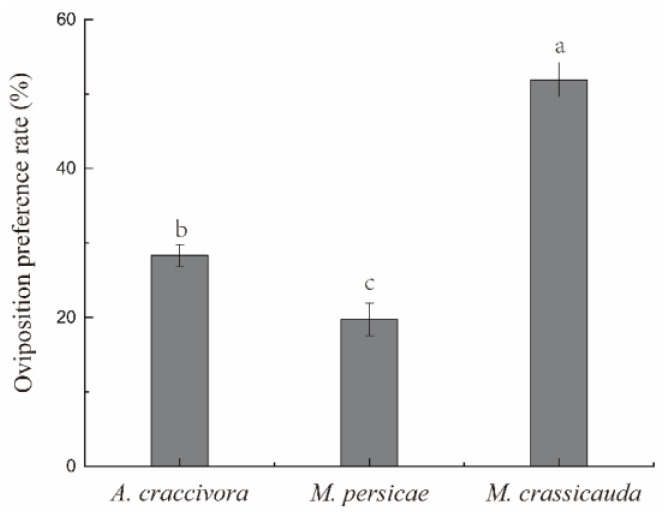
The oviposition preference rate of *E. balteatus* adults. Different lowercase letters in the figure reflect significant differences (*p* < 0.05).

**Table 1 insects-16-00774-t001:** Species and density of aphids preyed on by *E. balteatus* larvae.

Larvae of *E. balteatus*	Aphid Species	Density Settings
2nd instar	*A. craccivora*	10, 20, 30, 40, 50, 60, 70
	*M. persicae*	10, 20, 30, 40, 50, 60, 70
	*M. crassicauda*	5, 10, 15, 20, 25, 30, 35
3rd instar	*A. craccivora*	20, 40, 60, 80, 100, 120,140
	*M. persicae*	20, 40, 60, 80, 100, 120, 140
	*M. crassicauda*	10, 20, 30, 40, 50, 60, 70

**Table 2 insects-16-00774-t002:** The initial *E. balteatus*/aphid release ratios.

Aphid	Control Group	Hoverfly/Aphid
*A. craccivora*	1000	1:500, 1:1000, 1:2000, 1:4000, 1:6000
*M. persicae*	2000	1:2000, 1:4000, 1:6000, 1:8000, 1:10,000
*M. crassicauda*	250	1:250, 1:500, 1:1000, 1:1500, 1:2000

Control group indicates there contained only broad bean plantlets infested with aphids, without *E. balteatus* adult in the cage.

**Table 3 insects-16-00774-t003:** Different functional response models.

Name	Model	*z*	*Q*
Holling type II	*N_a_* = *aN*/(1 + *aNh*)	1	1
Holling type III	*N_a_* = *aN*^2^/(1 + *aN*^2^*h*)	2	1
θ-sigmoid	*N_a_* = *aN^θ^*/(1 + *aN^θ^h*)	*θ*	1
Beddington–DeAngelis	*N_a_* = *aN*/(1 + *γP* + *aNh*)	1	1 + *γP*
Arditi and Akçakaya	*N_a_* = *aNP*^−*m*^ /(1 + *aNP*^−*m*^*h*)	1	*P ^−m^*
Arditi and Ginzburg	*N_a_* = *aN/P*/(1 + *aNh/P*)	1	*P*

*N*: number of preys; *P*: number of predators; *a*: attack rate; *h*: handling time; *θ*, *γ*: parameters; *m*: interference coefficient. The term *Q* is either a constant or a function of *P*. In our study, since the number of predators is always 1, the Arditi and Akçakaya and Arditi and Ginzburg models become identical to the Holling type II model. Therefore, the four functional response models (Holling type II and III, θ-sigmoid, and Beddington–DeAngelis) were used for fitting in R version 4.3.1. According to Okuyama [26], the best model was selected using a general model selection method (Akaike information criterion (AIC)) [27]: AIC = log (SSQ_k_) + 2k/n, where *SSQ_k_* is the minimum sum of squares for the model with *k* parameters, and *n* is the number of trials. Differences in the oviposition preference rate, number of predatory, predatory preference, population decline rate, and number of larvae were analyzed using one-way analysis of variance, followed by Tukey’s post hoc test with proportional data first arcsine square root-transformed to meet the assumptions of normality and heteroscedasticity in SPSS version 25 (IBM, Armonk, NY, USA). Except for the functional response graphs, which were created in R version 4.3.1, all other graphs were completed using OriginPro 2021 (OriginLab Corporation, Northampton, MA, USA).

**Table 4 insects-16-00774-t004:** Parameters for the fits of four functional response models of 2nd- and 3rd-instar *E. balteatus* larvae to *A. craccivora*, *M. persicae*, and *M. crassicauda*.

Model	Attack Rate (*a*)	Handling Time (*h*)		SSQ	AIC
*A. craccivora*					
2nd instar					
Holling type II	1.927	0.028	Z = 1	335.934	2.640
Holling type III	0.154	0.036	Z = 2	253.395	**2.518**
θ-Sigmoid	0.097	0.036	Z = 2.179	251.719	2.572
Beddington– DeAngelis	1.93	0.028	*r =* 0.001	335.934	2.697
3rd instar					
Holling type II	1.353	0.014	Z = 1	1085.461	**3.149**
Holling type III	0.055	0.02	Z = 2	1210.885	3.197
θ-Sigmoid	0.739	0.016	Z = 1.195	1076.196	3.203
Beddington– DeAngelis	1.416	0.014	*r* = 0.046	1085.461	3.207
*M. persicae*					
2nd instar					
Holling type II	1.561	0.014	Z = 1	419.373	2.737
Holling type III	0.1	0.022	Z = 2	256.728	**2.524**
θ-Sigmoid	0.114	0.022	Z = 1.955	256.434	2.580
Beddington– DeAngelis	1.589	0.014	*r =* 0.018	419.373	2.794
3rd instar					
Holling type II	1.625	0.012	Z = 1	1754.829	3.358
Holling type III	0.06	0.016	Z = 2	1493.542	**3.288**
θ-Sigmoid	0.092	0.016	Z = 1.873	1488.689	3.344
Beddington– DeAngelis	1.658	0.012	*r* = 0.02	1754.829	3.416
*M. crassicauda*					
2nd instar					
Holling type II	1.694	0.051	Z = 1	77.833	2.005
Holling type III	0.265	0.068	Z = 2	60.272	**1.894**
θ-Sigmoid	0.258	0.068	Z = 2.015	60.269	1.951
Beddington– DeAngelis	1.7	0.051	*r =* 0.004	77.833	2.062
3rd instar					
Holling type II	1.275	0.021	Z = 1	328.515	2.631
Holling type III	0.087	0.032	Z = 2	323.501	**2.624**
θ-Sigmoid	0.339	0.028	Z = 1.508	302.637	2.652
Beddington– DeAngelis	1.293	0.021	*r* = 0.015	328.515	2.688

Values in boldface type indicate models with noteworthy support.

**Table 5 insects-16-00774-t005:** The population decline rate of *A. craccivora*, *M. persicae*, and *M. crassicauda* due to *E. balteatus* under different hoverfly/aphid ratios.

Hoverfly/Aphid	Population Decline Rate (%)			
	3rd Day	6th Day	9th Day	12th Day
*A. craccivora*				
CK	−179.00 ± 1.89c	−208.67 ± 6.43c	−378.67 ± 16.73c	−449.67 ± 27.86c
1:500	−186.67 ± 6.69c	−210.67 ± 8.13c	−18.00 ± 14.24a	94.67 ± 2.88a
1:1000	−176.67 ± 12.53c	−203.33 ± 6.43c	−66.66 ± 17.74b	100.00 ± 0.00a
1:2000	−104.00 ± 3.27b	−109.67 ± 3.65b	−36.50 ± 6.13ab	77.50 ± 5.18ab
1:4000	−108.67 ± 6.88b	−134.92 ± 7.33b	−93.58 ± 3.37b	35.58 ± 5.51ab
1:6000	−28.50 ± 2.75a	−42.44 ± 3.85a	−91.67 ± 7.20b	24.72 ± 4.61b
*M. persicae*				
CK	−46.83 ± 3.81bc	−59.50 ± 5.67bc	−131.5 ± 6.28b	−152.50 ± 2.62d
1:2000	−59.33 ± 6.95c	−71.33 ± 5.20c	1.33 ± 14.74a	96.67 ± 0.76a
1:4000	−17.67 ± 1.72a	−36.50 ± 0.93ab	16.42 ± 6.14a	95.42 ± 0.95a
1:6000	−19.61 ± 2.57a	−40.00 ± 4.35ab	20.67 ± 6.30a	60.72 ± 1.99b
1:8000	−25.54 ± 3.62ab	−22.83 ± 4.98a	−1.96 ± 4.16a	40.25 ± 5.31bc
1:10,000	−17.17 ± 2.76a	−27.23 ± 4.81a	−6.37 ± 7.83a	21.10 ± 7.02c
*M. crassicauda*				
CK	−34.67 ± 2.88ab	−108.00 ± 5.66abc	−244.00 ± 16.11c	−304.00 ± 21.75c
1:250	−25.33 ± 4.74a	−77.33 ± 10.38a	100.00 ± 0.00a	— —
1:500	−60.00 ± 6.53b	−182.00 ± 17.99c	−100.00 ± 36.48b	84.67 ± 7.56a
1:1000	−32.67 ± 1.90ab	−154.33 ± 22.29bc	−45.67 ± 14.32b	40.33 ± 16.98a
1:1500	−43.55 ± 8.41ab	−89.33 ± 7.10ab	−128.89 ± 4.28b	−98.22 ± 5.28b
1:2000	−52.83 ± 5.30ab	−102.00 ± 1.65ab	−128.67 ± 4.28b	−98.17 ± 5.27b

Values are means ± SE. Different lowercase letters in the same column reflect significant differences (*p* < 0.05).

**Table 6 insects-16-00774-t006:** The predatory preference of *E. balteatus* larvae among different aphid species.

Larva	Aphid Species	No. of Predatory	*Ci*
2nd instar	*A. craccivora*	9.33 ± 0.66a	0.04 ± 0.01a
	*M. persicae*	7.67 ± 0.34a	−0.06 ± 0.01b
	*M. crassicauda*	9.00 ± 0.58a	0.02 ± 0.01a
3rd instar	*A. craccivora*	16.67 ± 0.88a	0.17 ± 0.01a
	*M. persicae*	6.33 ± 0.34c	−0.30 ± 0.02c
	*M. crassicauda*	12.33 ± 0.66b	0.02 ± 0.01b

Values are mean ± SE. Different lowercase letters in the same column indicate significant differences (*p* < 0.05).

## Data Availability

The original contributions presented in this study are included in the article. Further inquiries can be directed to the corresponding author.

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
