# Peer review of "Evaluation on Biocontrol Efficacy of Episyrphus balteatus De Geer (Diptera: Syrphidae) Against Aphis craccivora, Myzus persicae, and Megoura crassicauda"

_insects, 2025, doi:10.3390/insects16080774_

Round 1
Reviewer 1 Report
Comments and Suggestions for Authors
Manuscript Review Comments
This is an interesting work presenting the role of Episyrphus balteatus in aphids control in China. Many observations have been carried out on this topic in Europe yet however in this manuscript there are some new information e.g. use of different functional response models.
The findings of this study are good for both farmers and the scientific community, but the manuscript needs major revision to upgrade and refocus the story for better understanding.
The English could be improved to more clearly express the research.
- Introduction
In my opinion the Introduction requires some improvement - introducing more literature related to the effectiveness of predatory hoverflies especially Episyrphus balteatus.
- In Introduction the authors suggest that hoverflies are excellent pollinators but do not provide any data to support this claim in their results. Please, remove this paragraph (line 44-61)
- Methods
The methodology is not clear as written. not all aspects of the work have been explained. It should be corrected and supplemented.
- line 116-122 - It is unclear how the study was conducted - whether aphids species were given separately or together into petri dishes,
- how the number of aphids eaten by larva was determined,
- whether only mature females of aphids or also larvae were given.
- why different number of aphids of each species were given .
- why different predator to prey ratios were used for different species of aphids (line 125-131).
- how long the experiment lasted, for how many days was voracity checked.
This is not clear. Please explain.
- Result
Figure 1 and 2 are poorly legible (not very clear)- please corrected them.
- Discussion
In Discussion the authors did not refer expressly to the tables and figures in Results at all. In discussion the authors usually explain the meaning of the own data and compare them with the data of other papers. However, the authors did not refer expressly to own data shown in Results.
- The discussion needs improvement. It contains a very poor number of publications on voracity of hoverflies (only 37 of which over 10 concern predatory species other than hoverflies).
- Information regarding the voracity of other predatory insects should be removed from the discussion – Lines 297-302, 313-315.
- The authors should supplement the list with manuscripts from all over the world concerning voracity of hoverflies especially Episyrphus balteatus. Many authors have worked on the effectiveness, of syrphids e.g.:
-Sunil Joshi, Chandish R. Ballal. Review Article Syrphid predators for biological control of aphids Journal of Biological Control, 2013, 27(3): 151–170.
-Tenhumberg, B. Estimating Predatory Efficiency of Episyrphus balteatus (Diptera: Syrphidae) in Cereal Fields. Environ. Entomol. 1995, 24, 687–691
-Kumar, A.; Kapoor, V.C.; Mahal, M.S. Feeding behavior and efficacy of three aphidophagous syrphids. J. Ins. Sci. 1996, 9, 15–18. 56.
-Agarwala, B.K.; Bhaumik, A.K.; Gilbert, F.S. Relative development and voracity of six species of aphidophagous syrphids in cruciferous crops. Proc. Ind. Acad. Sci. (Anim. Sci.) 1989, 98, 267–274
-Wojciechowicz-Żytko, E.; ˙ Dobiińska-Graczyk, M. Urban Green Space as a Reservoir of Predatory Syrphids (Diptera, Syrphidae) for Aphid Control in Cities. Agronomy 2025, 15, 953.
- References to other types of predatory insects are unnecessary.They should be removed from the list.
- The reviewer hopes that the authors state critically their opinions or enforcements on the data of other papers and that the authors refer to own data expressly and draw the conclusions.
- Conclusions - this chapter should be formulated
Formulate conclusions - give the main findings of the work. What did you find in your study and what is the implication for agriculture?
The manuscript will be ready for publication after major revision.
The English could be improved to more clearly express the research.
Reviewer 2 Report
Comments and Suggestions for Authors
The manuscript presents the results of an experiment on the efficacy of predation by E. balteatus on three different aphid species, common pests. The manuscript is well prepared, and the results are clearly presented. The results may be of quite broad importance for pest management and biocontrol. I have only a few small issues concerning the manuscript:
Line 46: should be: pupa
Line 123: Please indicate that in Table 1, column “larvae” concerns larvae of E. balteatus, and not aphids; aphids also have instars. Density means 10,20,30…60 of adult aphids per petri dish?
Line 259: Please, citing results values, refer to the proper Table with results
Line 272: remove the dot after the brackets and put the species name into italics
Perhaps it would be good to indicate what flowering plants could attract E. balteatus towards crops. Such information could add some pragmatic value to the manuscript.
Round 2
Reviewer 1 Report
Comments and Suggestions for Authors
Dear Authors-The figures (Fig.1, 2) have been corrected, however the tables placed inside the figures are still difficult to read - it is suggested that they be enlarged and also the lines on the charts should be thicker.